# The Effect of Shear Rate on Dynamic Gelation of Phenol Formaldehyde Resin Gel in Porous Media

**DOI:** 10.3390/gels8030185

**Published:** 2022-03-17

**Authors:** Haiyang Yu, Zhenfu Ma, Lin Tang, Yuesheng Li, Xianzhen Shao, Yunxia Tian, Jun Qian, Jing Fu, Dong Li, Long Wang, Fengguo Ren

**Affiliations:** 1College of Energy and Mining Engineering, Shandong University of Science and Technology, Qingdao 266590, China; 2Shengli Oilfield, SINOPEC, Dongying 257200, China; mazhenfu.slyt@sinopec.com (Z.M.); tanglin820.slyt@sinopec.com (L.T.); liyuesheng.slyt@sinopec.com (Y.L.); shaoxianzhen.slyt@sinopec.com (X.S.); tianyunxia.slyt@sinopec.com (Y.T.); qianjun132.slyt@sinopec.com (J.Q.); fujing175.slyt@sinopec.com (J.F.); lidong150.slyt@sinopec.com (D.L.); wanglong.slyt@sinopec.com (L.W.); renfengguo.slyt@sinopec.com (F.R.)

**Keywords:** shear rate, dynamic gelation, phenol formaldehyde resin gel, gelation time, gel strength, porous media, water flooding, injected speed

## Abstract

Polymer gel is the most widely used plugging agent in profile control, whose formula and injected speed are very important process parameters. It is very significant to study the effect of shear rates on the dynamic gelation of polymer gel in porous media for selecting suitable formula and injection speed. Taking the phenol formaldehyde resin gel with static gelation time of 21 h in ampoule bottle as research objective, it was studied the dynamic gelation process and subsequent water flooding in porous media under different injected speeds by a circulated equipment. The results shown that final dynamic gelation time is 2.4 times longer than the static gelation time in porous media. The gel particles are formed and mainly accumulated in the near wellbore zone after dynamic gelation. Injection speed has little effect on the dynamic gelation time in porous media, but has a great effect on the gel strength. The effect of injection speed on dynamic gel strength is evaluated by established the quantitative relationship between shear rate and dynamic gel strength. According to subsequent water flooding results, gel particles have certain plugging capacity in the near wellbore zone. The plugging ability declines obviously with an increasing injection speed. The experimental results provide theoretical support for the successful application of polymer gel used in profile control.

## 1. Introduction

Profile control and water plugging is a very important technology to reduce water cut and improve oil recovery [1,2,3]. Polymer gel is one of the most commonly used and important agents, which is widely used in oil fields and has achieved remarkable results [4]. Gelation time and gel strength are the most important performance indicators of polymer gel [5]. The recent research results on the dynamic gelation process in porous media show that the dynamic gelation process can be divided into three stages: induction stage, cross-linking stage and stabilization stage [6,7]. However, the gelled liquid is sheared when it passes through the rock pore throat, which results in significant changes in dynamic gelation time, gel strength and gel speciation. Technically, the faster the injected speed is, the more significant the shear effect of rock is. So it is important to study the effect of injection speed on the dynamic gelation of polymer gel in porous media.

Different experimental methods have been used to study the dynamic gelation of polymer gelation. Seright used a 100 foot long iron tube (0.03 inch inside diameter) to simulate the flow experiment of polymer gel in the crack [8]. The results showed that the residual resistance coefficient reached the highest point at 20 feet and began to decrease, while the residual resistance coefficient at the middle and rear end (60–100 feet) of the iron tube remained unchanged. Stan McCool used 1036 feet long stainless steel conduit (0.0566 inch inner diameter) to simulate fractured formation to study the dynamic gelation behavior of chromium acetate gel in the process of porous media [9]. Wu used a 32 m long sand pack to conduct the core flowing experiments before and after gelation [10]. He studied gelation behavior of resorcinol-Hexamethylenetetramine-HPAM gel in bulk and porous media, showing that the gelation could occur during flow in porous media and the gelation location could be inferred to determine gel placement [11]. The author studied the dynamic gelation process of polymer gel in porous media in the early stage [12,13,14]. The results showed that shear had almost no influence on gel strength during the induction stage but in the cross-linking stage, shear could degrade gel strength sharply. Moreover, there are many studies on the dynamic cross-linking mechanism of polymer gel. Allainc [15] and klaveness [16] study the cross-linking reaction process of chrome gel by UV-vis spectrophotometry. Jain determined the reaction mechanisms of chromium acetate crosslinking system by a rheometer [17]. Czakkel evaluated dynamics and structure during formation of a cross-linking polymer gel by in situ X-ray photon correlation spectroscopy and small-angle X-ray scattering techniques [18]. Ryu investigated the dynamic behavior of imogolite-poly (acrylic acid) nanocomposite hydrogels using dynamic light scattering [19]. Besides, there are other experimental methods to study the dynamic gelation of polymer gels [20,21,22]. These studies are focused on studying whether the polymer gel can be gelled in the flow process and investigating the dynamic cross-linking mechanisms, but the influence of injection speed on dynamic gelation is not made entirely clear.

The injection speeds affect the shear strength of polymer gel during dynamic gelation in porous media. Based on the most widely used phenolic resin gel, we study the relationship between static gelation time and dynamic gelation time, the effect of injection speed on dynamic gelation in porous media through establishing the quantitative relationship between injection speed and shear rate, the gel strength of dynamic gelation under different injection speeds by analyzing the residual resistance coefficient of subsequent water flooding The results provide theoretical basis for the selection of optimal injection speed for the field application of polymer gel.

## 2. Results and Discussion

### 2.1. Results

#### 2.1.1. Static Gelation in Ampoule Bottle

The result of the static gelation in ampoule bottle is shown in Figure 1. With an increasing time, the viscosity of the system keeps basically unchanged firstly, then increases rapidly, and finally tends to be stable. The gelation process of polymer gel experienced longer periods of induction stage, cross-linking stage and stabilization stage [6,7]. According to the research results of Mehdi mokhtari, the gelation time is divided into initial gelation time (IGT) and final gelation time (FGT). The initial gelation time refers to the time when the viscosity of the system starts to rise obviously at the beginning of the cross-linking reaction, which is the boundary point between the slow induction stage and the fast cross-linking stage. The final gelation time is the time when the viscosity of the system reaches a stable level at the end of the cross-linking reaction, which is the dividing point between fast gelation stage and the strength stabilization stage. The IGT and FGT of the gel system studied in this work are 12 h and 21 h, respectively. And the stable viscosity is 22,386 mPa·s.

Based on the SEM results, the polymer gel formed regular network structure after static gelation in ampoule bottle, which is formed by the cross-linking of amido group in polymer and hydroxymethyl group in phenolic resin. In the network structure, there are several pores with similar sizes falling into the range of 3 μm~5 μm, which are bound water covered by the network structure of the gel system.

#### 2.1.2. Static Gelation in Porous Media

The result of the static gelation in porous media is shown in Figure 2. With time increasing, the RRF keeps basically unchanged firstly, then increases rapidly, and finally tends to be stable. That is similar to the static gelation in ampoule bottle. In the induction stage, the amide group of the polymer molecule reacts with the hydroxymethyl group of the phenolic resin molecule to form independent structural units. But these units have not been effectively connected to each other and the viscosity of the system did not increase sharply. The blocking effect of gel has not been systematically developed. In the cross-linking stage, the RRF increases fast. The formed structural units appear the form of aggregates. The aggregates grow in different directions. They gradually form a network structure, which adsorbs on the rock surface or exists between the sand throats. Until the end of the reactions, the structural units are completely cross-linked and the RRF remains unchanged. The IGT and FGT of the used gel system are 17 h and 40 h, respectively. And the stable RRF is 48. 

Compared to static gelation in ampoule bottle, the IGT of static gelation in porous media is 1.5 times longer and the FGT is 2 times longer. In the process of polymer gel injection into sand packs, polymer molecules and cross-linker molecules are sheared by sand throats. Due to the different molecular weight of polymer and cross-linker, the migration speed in porous media is different. They are separated gradually, which changes the ratio of polymer and cross-linker, thus prolonging the gelation time.

#### 2.1.3. Dynamic Gelation in Porous Media under Different Injected Speeds

The results of dynamic gelation in porous media under different injection speeds are shown in Figure 3. With the increase of time, the pressure difference ΔP_ad_ of dynamic gelation in porous media is basically unchanged, then increases rapidly, and finally tends to be stable. It shows that during the dynamic gelation process, the gel system experiences the induction stage, the gelation stage and the stability stage, similar to what happens during the static gelation process. But in the fast gelation stage, there is a phenomenon of “zigzag pressure increase”, different to what happens during the static gelation. That may be due to the shear action by porous media. At different injection speeds, the ΔP_bd_ increases in varying degrees, but there is a significant time delay phenomenon. The results show that the migration of the used gel can occur during the dynamic gelation in porous media and have the plugging ability in the deep formation.

There are two forces acting on the polymer gel during the dynamic gelation process in porous media; one is the intermolecular chemical cross-linking force, and the other is the shear force of porous media. Under the shear action of porous media, when the cohesive structure of gel is overcome by shear force, the gel aggregate is sheared to form a dispersed gel particle system instead of a bulk gel, as shown in Figure 4. Gel particles mainly accumulate at the small pore throats or adsorb on the surface of porous media to reduce the seepage channel of porous media. Compared to the static gelation in porous media, we cannot observe the network structure of the gel system after dynamic gelation in porous media.

The results of IGT and FGT of dynamic gelation in porous media are shown in Table 1. And under different injection speeds, the IGTs are in the range of 22 h to 25 h and the FGTs are in the range of 93 h to 98 h. This shows that the injection speed has little effect on the dynamic gelation time in porous media. Compared to static gelation in porous media, the IGT of dynamic gelation in porous media is 1.4 times longer and the FGT is 2.4 times longer. This may be due to the shear force of porous media and chromatographic separation effect.

#### 2.1.4. Water Flooding after Dynamic Gelation in Porous Media

When the dynamic gelation process is over, the subsequent water flooding is carried out to evaluate the plugging ability of the used gel after dynamic gelation. The change of pressure difference of water flooding with pore volume (V_p_) under different injection speeds is shown in Figure 5. Under different injection speeds, the variation trends of pressure differences in the subsequent water flooding are similar. First, they increases rapidly to the maximum value, then decrease, and finally tend to be stable. With an increasing injection speed in the process of dynamic gelation in porous media, the pressure differences during the subsequent water flooding show a declining trend. Moreover, the pressure difference of the ad section is significantly greater than that of the bd section. This indicates that the location of gel plugging after dynamic gelling is still the near wellbore section.

The RRF of the ad section and the bd section are calculated based on the stablized pressure differences. And the result is shown in Figure 6. With injection speed increasing, the RRF of the subsequent water flooding decreases. This may be because, with the increase of injection rate, the shear rate of porous media increases, and the net structure damage of polymer gel is more serious. The plugging ability of the used gel after dynamic gelation could become weaker. Moreover, The RRF of the ad section is bigger obviously than that of the bd section. 

Compared to the static gelation in porous media, the RRF of static gelation is 48, when the injection speed is lower than 1 mL/min, the RRF of dynamic gelation is bigger than that of static gelation. But when the injected speed is greater than 1mL/min, the RRF of dynamic gelation is smaller than that of static gelation. In the process of dynamic gelation, the polymer gel network structure is destroyed by shear force, and the plugging ability is reduced. However, when the injected speed is low, more gel particles would accumulate in the near well section, and the plugging capacity may be greater than that of static gelation.

### 2.2. Discussion

#### 2.2.1. Gelation Time of Static and Dynamic

In the static gelation process, polymer gel is only affected by the cross-linking force, so the gelling unit gradually increases, and finally forms a three-dimensional network structure. However, in the dynamic gelation of porous media, the polymer gel is affected by the cross-linking force which is conducive to the formation of the network structure, and the shear force which destroys the network structure. The IGT and FGT is prolonged and the plugging capacity becomes weaker with injection speed increasing. During the dynamic gelation process, the used gel experiences the induction stage, the gelation stage and the stability stage. In the induction stage, the polymer molecules are cross-linked by the cross-linker molecules to form gelling units. The viscosity does not change significantly and the shear force has little effect on the viscosity. However, in the gelation stage, the gelling units aggregate to form a network structure, which is greatly affected by the shear force. Therefore, in the actual application of polymer gel for profile control, the gelation time of polymer gel should refer to the initial gelation time (IGT) of dynamic gelation in porous media.

#### 2.2.2. The Relationship between Shear Rate and Viscosity of Dynamic Gelation

When the pore structure is similar, the greater the injection rate, the greater the shear force. And the shear rate can be calculated by Equation (1) [23]. The rheological index of used gel is determined according to the relationship between shear stress and shear rate. The shear rates under different injected speeds are listed in Table 2.
(1)$$\gamma =\frac{3n+1}{n}\times \frac{\nu }{\sqrt[{0.5}]{8C^{\prime }K\Phi }}$$

Equation (1) predicts the shear rate in porous media. *γ*—shear rate, s^−1^; n—Rheological index, MPa·s^n^; *V*—injected speed, mL/min; *C*′—tortuosity, percentage; *K*—permeability, μm^2^; *Φ*—porosity, percentage.

ΔP_ad_ can be used to calculate the viscosity of the used gel in the dynamic gelation process in porous media. The relationship between viscosity and shear rate can be established according to the viscosity and calculated shear rate after dynamic gelation, shown in Figure 7. With an increase in the shear rate, the viscosity of dynamic gelation decreases. When the permeability is 5–8 μm^2^ and the shear rate is more than 10 s^−1^, the viscosity of the gel after dynamic gelation is low. The same conclusion can be drawn from the results of subsequent water flooding experiments.

The relationship between viscosity and shear rate satisfies the formula (y = a × x^b^). a value represents the viscosity value after dynamic gelation. The larger the a value is, the greater the viscosity of the gel system is. The b value reflects the influence of shear rate on the gel system. The larger the b value is, the greater the influence of shear rate on the gel system is. And the gel system may have the worse shear resistance. Comparing with the actual data and the fitting curve, we can see that when the shear rate is small, the variation range of the pressure difference in the stable stage is gentle, and the difference between the actual value and the fitting value is not significant. However, with the increase of injection rate, the fitting value is larger than the actual value, which is caused by the experimental method of cyclic injection. It can be seen from the above experiments that the injected speed has a great influence on the dynamic gelation in porous media. When the pore structure is similar, with the increase of injection speed, the shear rate increases, the viscosity of gel after dynamic gelling is smaller, and the plugging capacity of subsequent water flooding is weaker. Therefore, the effect of injection rate should be considered in the practical application of polymer gel, and the injection rate should be reduced as much as possible to obtain a higher gel viscosity when the field conditions allow.

The relationship between injection speed and shear rate under different permeabilities can be established by formula in Figure 7, as shown in Figure 8. When the permeability is constant, the shear rate of polymer gel in porous media increases with the increase of injection speed. At a certain injection speed, the shear rate decreases with the increase of permeability. The results of dynamic gelation experiment show that the viscosity of the used gel is lower when the shear rate is more than 10 s^−1^. When the shear rate is less than 10 s^−1^, the selection range of injection speed increases with the increase of permeability. According to Figure 8, the injection velocity range with a higher viscosity value after dynamic gelation of polymer gel under different permeabilities can be obtained. In the field application of polymer gel, the injected speed is required stricter with reservior permeability decrease.

## 3. Conclusions

Taking the polymer gel composed of 0.2 wt% HPAM and 0.6 wt% PFR as the research object, we study the dynamic gelation process in porous media with different injection speeds and the consideration of the subsequent water flooding. The main conclusions are as follows:Compared to static gelation in porous media, the initial gelation time (IGT) and final gelation time (FGT) of dynamic gelation in porous media are respectively 1.4 times and 2.4 times longer. The gel particles will be formed in the process of dynamic gelation, mainly accumulating in the near wellbore zone. And the bulk network structure will be formed under static gelation.Injection speed has little effect on the dynamic gelation time in porous media, but has a great effect on the gel strength. The relationship between viscosity and shear rate satisfies the formula (y = a × x^b^). With the increase of injection speed, the shear rate increases, and the gel strength after dynamic gelation is weaker.The results of the subsequent water flooding show that after dynamic gelation in porous media, gel particles have certain plugging capacity in the near wellbore zone. With the increase of injection speed, the plugging ability shows a declining trend.Based on the research results, in filed application of polymer gel, we need to consider the initial gelation time (IGT) of dynamic gelation to choose the suitable gel formula and make the injection speed as small as possible.

## 4. Materials and Methods

### 4.1. Materials

The polymer employed in this research is the classical partially hydrolyzed polyacrylamide (HPAM), whose molecular weight is 1.2 × 10^7^ and the degree of hydrolysis is 22%. The cross-linker named phenol formaldehyde resin (PFR) is synthesized by phenol and formaldehyde under alkaline condition. The used synthetic water (SW) contain 6921 ppm Na^+^, 412 ppm Ca^2+^, 148 ppm Mg^2+^ and 11,853 ppm Cl^−^. The gel system is composed of 0.2 wt% HPAM and 0.6 wt% PFR. And the following experiments are conducted under 75 °C. 

### 4.2. Methods

#### 4.2.1. Static Gelation in Ampoule Bottle and Porous Media

The static gelation of the used gel system in ampoule bottle could form a bulk three-dimensional network structure. And the gelation time and gel strength are confirmed by the change of viscosity with time as measured by rheometer.

The static gelation of the used gel system in porous media could form a three-dimensional network structure adsorbed on porous media surface. And the gelation time and gel strength are confirmed by coefficient of residual resistance (RRF) with time by the core flooding experiments. RRF experiments are operated with sand packs with 2.50 cm ID and 10.00 cm length fitted by clean micro glass beads with different mesh counts and saturated by SW. We first prepare some sand packs with the permeability at approximately 2.5 µm^2^. After injecting one pore volume (V_P_) gelation fluid, we set the system temperature at a constant temperature of 75 °C. We finally measure the RRF during water flooding under the injected speed of 1 mL/min. 

#### 4.2.2. Dynamic Gelation in Porous Media under Different Injected Speeds

The dynamic gelation of the used gel system in porous media under different injection speeds are operated with sand pack with 2.50 cm ID and 100.00 cm length fitted by clean micro glass beads with different mesh counts and saturated by SW. So dynamic gelation of HPAM/PFR gel in porous media could be estimated with the circulating device, which was comprised of two piston containers at room temperature and a sand pack in 75 °C (See Figure 9). First, one PV gelant in container I was injected into the sand pack. Then we connect the outlet of sand pack with container II and adjust temperature to 75 °C. One PV gelant in container II was injected into sand pack and initial one PV gelant in sand pack was displaced into container I again. By adjusting the valves, the one PV gelant in container I was injected into sand pack, and the one PV gelant in sand pack was displaced into container II again. The two PV gelant was alternately injected into the sand pack with 2.50 cm ID and 100.00 cm length. There are two internal pressure taps 30 and 70 cm from the inlet, shown in Figure 1. In the whole process, we record the change of injected pressure drops with time. And the gelation time and gel strength are confirmed by the curve of pressure difference ΔP_ad_ with time. The processes of dynamic gelation under different injection speeds are operated by changing the pumping rate from 0.125 mL/min to 2 mL/min. 

#### 4.2.3. Water Flooding after Dynamic Gelation in Porous Media

In order to confirm the RRF of the used gel system after dynamic gelation in porous media, the water flooding with the injected speed of 1 mL/min is conducted when the process of dynamic gelation is finished. And the gel strength of dynamic gelation can be determined by the curve of the pressure difference with injected pore volume of water.

## Figures and Tables

**Figure 1 gels-08-00185-f001:**
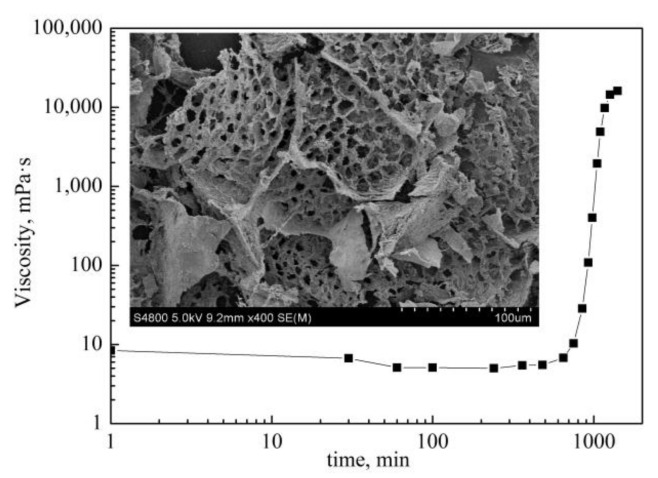
The curve of viscosity with time under static gelation in ampoule bottle.

**Figure 2 gels-08-00185-f002:**
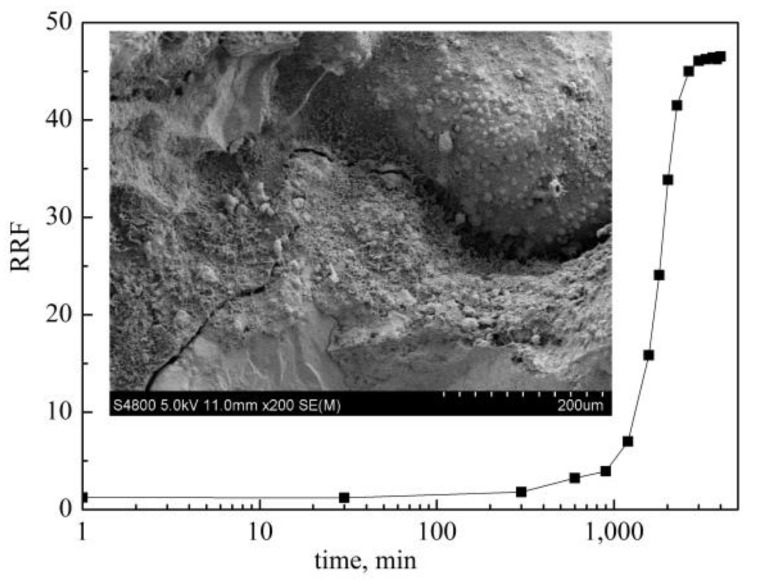
The curve of RRF vs. time under static gelation in porous media.

**Figure 3 gels-08-00185-f003:**
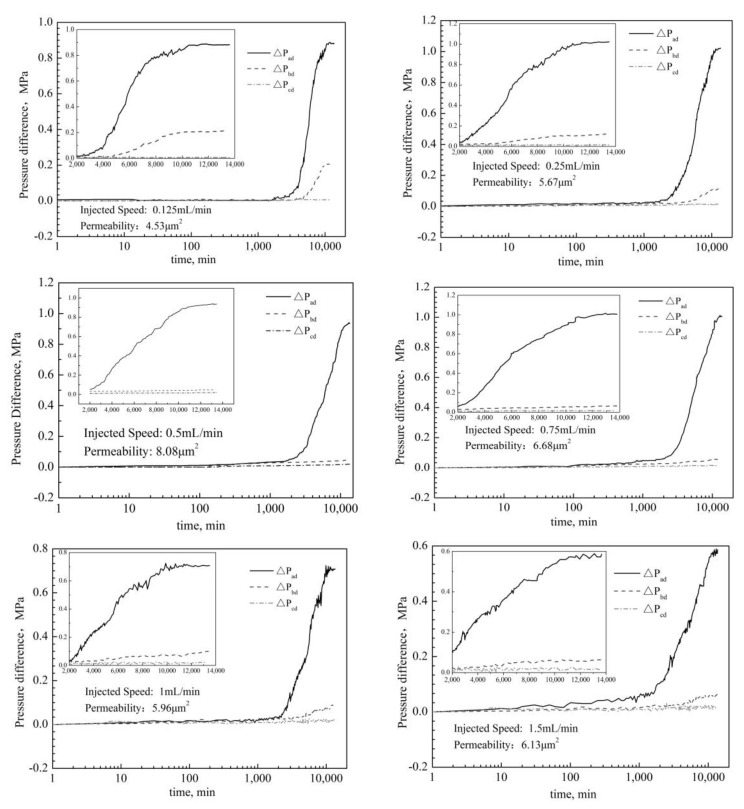
The curve of pressure difference vs. time under dynamic gelation in porous media.

**Figure 4 gels-08-00185-f004:**
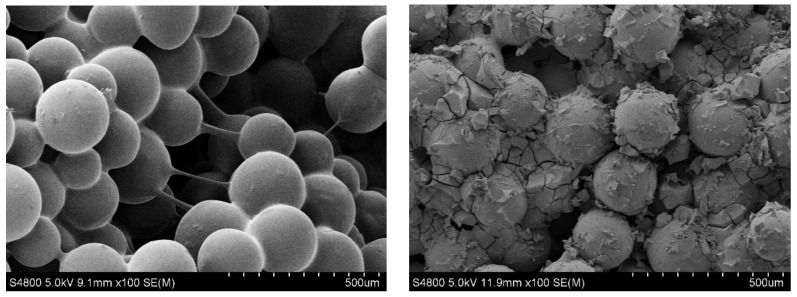
The micro morphology of porous media before and after dynamic gelation of PFR gel.

**Figure 5 gels-08-00185-f005:**
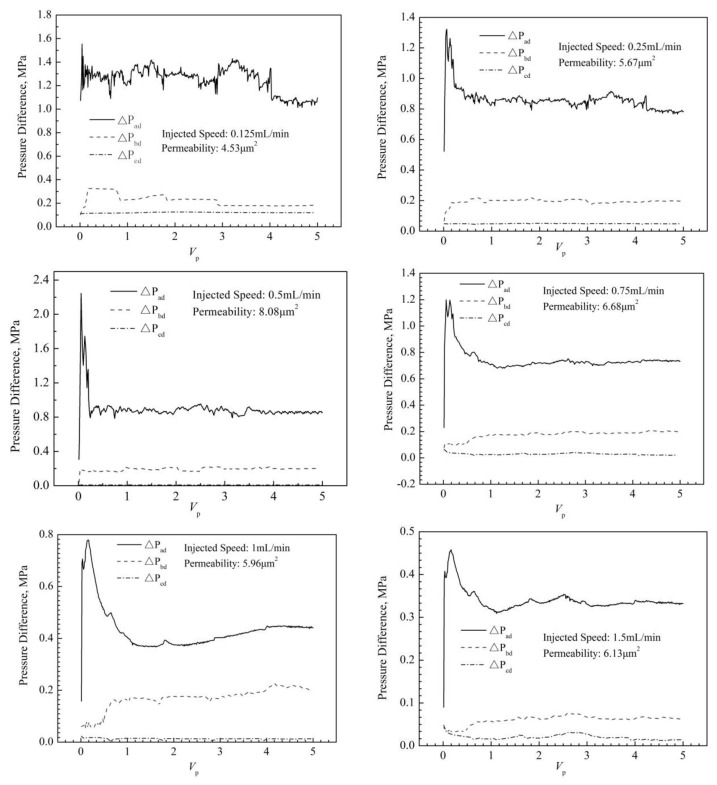
The change of subsequent water flooding pressure difference with injected pore volume.

**Figure 6 gels-08-00185-f006:**
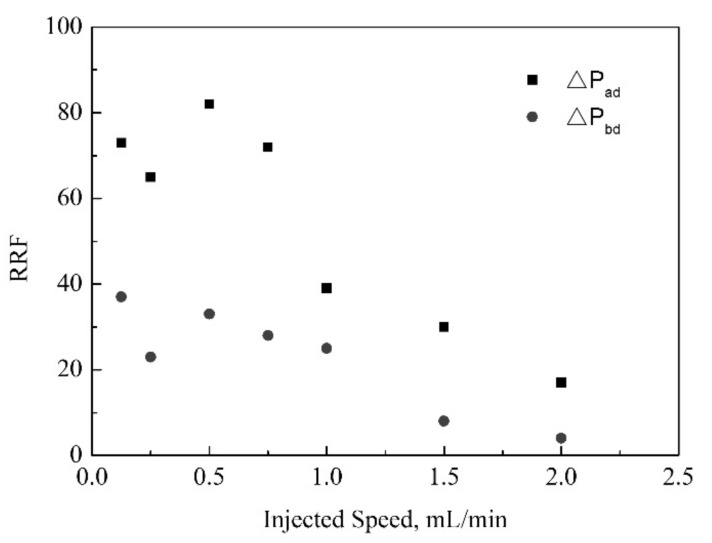
The change of RRF during the subsequent water flooding vs. the injection speed.

**Figure 7 gels-08-00185-f007:**
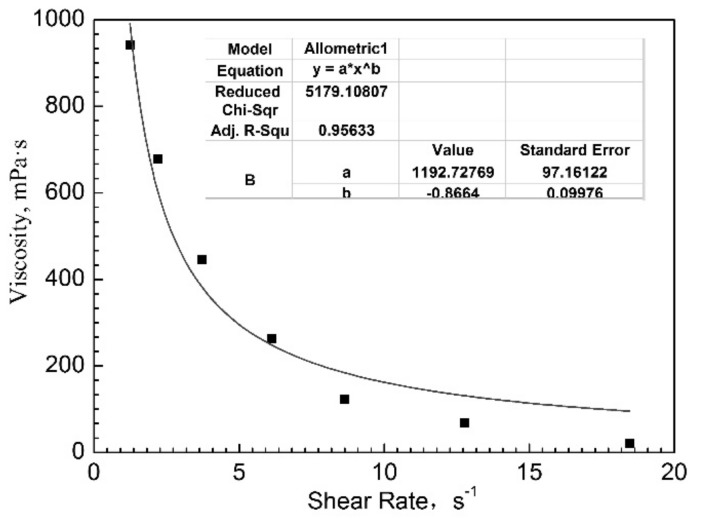
The change of viscosity of dynamic gelation in porous media with shear rate.

**Figure 8 gels-08-00185-f008:**
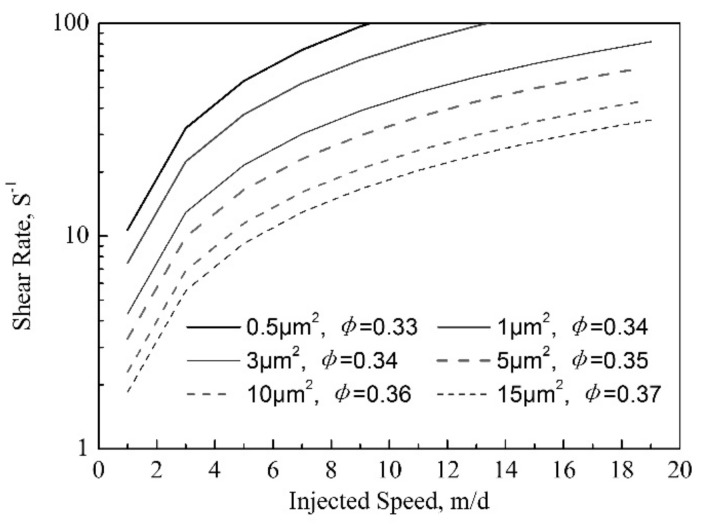
The relationship between injection speed and shear rate under different permeabilities.

**Figure 9 gels-08-00185-f009:**
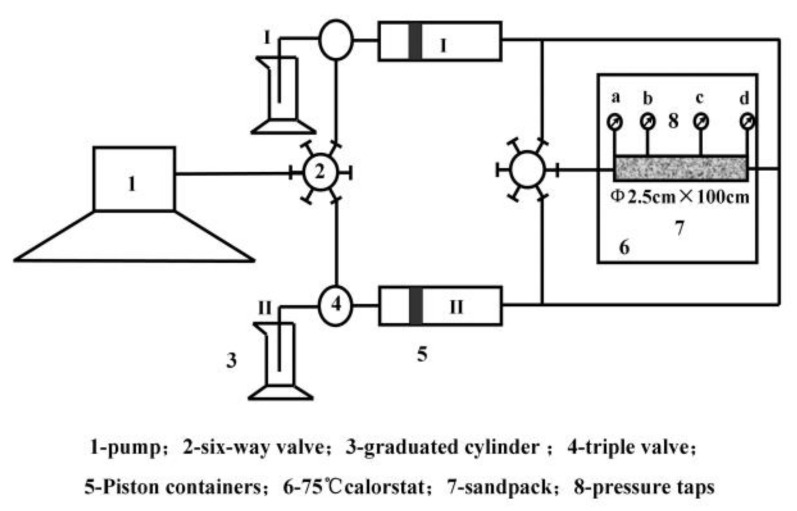
Schematic of circulated equipment for dynamic gelation in porous media.

**Table 1 gels-08-00185-t001:** Gelation times of dynamic gelation in porous media under different injection speeds.

Code	Injected Speed, mL/min	Permeability, μm^2^	Dynamic Gelation, h	
			IGT	FGT
1	0.125	4.53	25	93
2	0.25	5.67	23	94
3	0.5	8.08	22	95
4	0.75	6.68	23	96
5	1	5.96	24	96
6	1.5	6.13	24	97
7	2	5.22	25	98

**Table 2 gels-08-00185-t002:** Gelation times of dynamic gelation in porous media under different injection speeds.

Code	Injected Speeed, mL/min	K, μm^2^	Porosity	n	Tortuosity	Shear Rate, s^−1^
1	0.125	4.53	0.357	0.440	2.29	1.26
2	0.25	5.67	0.363			2.23
3	0.5	8.08	0.367			3.71
4	0.75	6.68	0.379			6.02
5	1	5.96	0.361			8.71
6	1.5	6.13	0.364			12.83
7	2	5.22	0.358			18.69

## Data Availability

The study did not report any data.

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
