# Peer review of "The Effect of Shear Rate on Dynamic Gelation of Phenol Formaldehyde Resin Gel in Porous Media"

_gels, 2022, doi:10.3390/gels8030185_

Round 1
Reviewer 1 Report
In general, the paper proposed by Authors is worth considering for publication in Gels. The main concept of presented research is interesting. Nonetheless, some revisions are strongly suggested, i.e.:
1) Abstract of the paper should be more specific and present more quantified data.
2) From editorial viewpoint, it should be a pause between a word and a reference in brackets - it should be corrected in the whole paper.
3) Quality of Figure 3. should be significantly improved because now it is poorly legible. Moreover, all subfigures should be noted as a), b) etc. and described in more detail. The same applies to Figure 5. The rest of figures should also be improved to be more legible and clearer.
4) Discussion over the results of performed studies should be supported to a more extent and compared to the results of other similar works.
5) Section References should be corrected and prepared according to the requirements of the Journal. The whole journal names should be replaced by their abbreviations.
6) Language of the paper should be significantly improved.
Author Response
Thank you for the careful review.

Reviewer 2 Report
The authors studied the relationship between static gelation time and dynamic gelation time, the effect of injected speed on dynamic gelation in porous media through establishing the quantitative relationship between injected speed and shear rate, the gel strength of dynamic gelation under different injected speeds by analyzing the residual resistance coefficient of subsequent water flooding. This reviewer recommends that this manuscript be published in this journal "Gels" because the results provided in this paper are interesting and are of great interest to the readers of "Gels".
Author Response
Thank you very much for giving me such a high evaluation. Your approval is the greatest encouragement to my growth.

Reviewer 3 Report
The article can be accepted for publication after addressing some suggestions.
Please improve the quality (resolution) of Figure 1 and Figure 3.
Line 17: please replace "is" for "was"
Please revise thoroughly the text, there are several typos in the manuscript, For example, in Line 79. figure.1., should be in my opinion, Figure 1. The same for tables should be Table 1. It may be wordy start sentences with And...... please revise and improve.
Line 90: Please revise the sentence
In my opinion, there are plenty of open sentences reporting results. For instance, lines 110 to 113; 117 to 121; 128-129; ...... as a suggestion, can be improved or merged into single paragraphs.
Author Response
Thank you very much for my paper acceptation. And I have completed the modification of figures as required.
